# Applying a Capabilities Approach to Understanding Older LGBT People’s Disclosures of Identity in Community Primary Care

**DOI:** 10.3390/ijerph17207614

**Published:** 2020-10-19

**Authors:** Michael Toze, Julie Fish, Trish Hafford-Letchfield, Kathryn Almack

**Affiliations:** 1Lincoln Medical School, University of Lincoln, Brayford Pool, Lincoln LN6 7TS, UK; 2School of Applied Social Sciences, De Montfort University, Leicester LE1 9BH, UK; jfish@dmu.ac.uk; 3School of Social Work and Policy, University of Strathclyde, Glasgow G4 0LT, UK; trish.hafford-letchfield@strath.ac.uk; 4School of Health and Social Work, University of Hertfordshire, Hatfield AL10 9AB, UK; k.almack@herts.ac.uk

**Keywords:** sexuality, gender identity, primary care, health capabilities

## Abstract

Internationally, there is increasing recognition that lesbian, gay, bisexual and trans (LGBT) populations experience substantial public health inequalities and require interventions to address these inequalities, yet data on this population is often not routinely collected. This paper considers the case study of the UK, where there are proposals to improve government and health data collection on LGBT populations, but also a degree of apparent uncertainty over the purpose and relevance of information about LGBT status in healthcare. This paper applies a health capabilities framework, arguing that the value of health information about LGBT status should be assessed according to whether it improves LGBT people’s capability to achieve good health. We draw upon 36 older LGBT people’s qualitative accounts of disclosing LGBT status within UK general practice healthcare. Participants’ accounts of the benefits and risks of disclosure could be mapped against multiple domains of capability, including those that closely align with biomedical accounts (e.g., longevity and physical health), but also more holistic considerations (e.g., emotion and affiliation). However, across all domains, individuals tend to assess capabilities at an individual level, with relatively little reference to population-level impact of disclosure. Clearer articulation of the benefits of disclosure and data collection for the collective capabilities of LGBT populations may be a beneficial strategy for improving the quality of information on LGBT populations.

## 1. Introduction

Internationally, there has been growing recognition that sexual orientation and gender identity is associated with inequalities within public health, and that this requires proactive action by governments and international agencies [1,2,3,4]. Public health relies on the availability of large-scale population data in order to assess the needs of populations and the impact of interventions. However sexual orientation and gender identity are often not consistently recorded in key datasets such as censuses, patient records and mortality registries. This has resulted in a lack of large-scale data on key later-life morbidity and mortality outcomes in LGBT populations, such as cardiovascular disease, diabetes, cancer and other long-term conditions and conditions of ageing [5,6,7,8]. There are a number of social and practical factors that may underpin this historic lack of data collection, including the small, dispersed nature of the population; the history of criminalisation and pathologisation; concerns about sensitivity and privacy; uncertainty over when and how to record data; and a perception that LGBT populations have few or no distinct health needs [9,10,11,12].

In the UK, the Office for National Statistics now proposes to ask about sexual orientation and gender identity for the first time within the 2021 census, with consultation indicating that academics and health providers saw data collection as valuable for meeting the needs of LGBT populations [13]. However, particularly for gender identity, there have been ongoing challenges and controversies in developing a question that is clear for all users, sensitive towards gender-diverse respondents, and that maintains comparability with earlier data collections on sex [14,15].

Separate to the proposals for the Census, the National Health Service (NHS) England recently introduced a standard for sexual orientation monitoring in healthcare (data on gender identity is being separately considered), establishing for the first time a standard recording format that permits comparison and cross-reference between different recording systems within the English NHS [16]. However, the accompanying guidance states that the standard does not mandate health bodies to collect data on sexual orientation, nor is it directive as to how data should be recorded and processed. The guidance also states that sexual orientation data should not be linked to individual patient records where the data “has no purpose in relation to” individual assessment or treatment [17]. The guidance therefore does not support the public health case for collecting data in order to proactively understand and address the needs of LGBT populations. It further assumes that it will always be objectively clear to both patient and clinician whether or not discussing and recording LGBT identity has a purpose, and that this purpose must be medical in nature. However, some LGBT people may want their sexuality or gender identity recorded for other reasons, for example because it is an important part of their identity [18].

Brooks et al., reviewing the literature on disclosures of LGBT identity within healthcare, identified that perceptions of relevance to health was one of the most frequently identified considerations regarding disclosure [19]. Similarly, in a recent UK Government Equalities Office survey on LGBT needs, 84% of respondents cited lack of relevance as the reason for not disclosing that they were LGBT within healthcare [20]. However, Fish argues that ‘relevance’ is defined in heterosexist (and by extension, cissexist) terms: LGBT identity is most likely to be considered relevant in situations in which heterosexuality is actively and explicitly presumed, particularly reproductive and sexual healthcare [11]. LGBT health is therefore framed reductively, within a biomedical understanding of health. For older LGBT people, the combination of this perception that LGBT health is a sexual health matter, along with the cultural assumption that older people are desexualised may further heighten assumptions that being LGBT has little significance in later-life healthcare [21,22]. However, interviews with LGB people themselves point to various understandings of interactions between sexual orientation and health: while some individuals do focus upon a biomedical account of relevance, others emphasise that disclosing details of self can promote holistic wellbeing through factors such as sense of personal authenticity or recognition of key emotional relationships and social networks [18,23].

Biomedical conceptualisations of relevance to health may be particularly limiting in general practice, given that the strengths of general practice are widely considered to lie in its ability to develop an ongoing patient relationship, manage continuing care, understand the patient’s social context, and undertake preventative and early interventions that may minimise the need to seek specialised care [24]. Prevention and early intervention has been emphasised in NHS England’s strategic planning [25,26], suggesting that holistic discussion of patient circumstances—including information which does not have immediate treatment utility, but could have value for planning preventative care—should be seen as important. Nonetheless, the guidance on the new sexual orientation standard suggests that at present, relevance of LGBT status is still understood narrowly [17]. This paper seeks to reconsider the ongoing issue of relevance by applying a conceptual approach commonly used in health inequalities research: the health capabilities framework.

### The Health Capabilities Approach

The health capabilities approach suggests that public health and wellbeing interventions should be evaluated according to what people can do, rather than what people actually do [27]. For example, a nutrition intervention should be assessed on whether it improves capability to select and obtain a healthy diet, rather than on whether individuals actually change their eating patterns. It is suggested that this helps to resolve traditional public health dilemmas between paternalism and autonomy through focusing on the creation of a social context within which all members of society are capable of pursuing and achieving good health, while also retaining individual freedoms to make alternative choices [27,28,29]. It therefore encourages consideration of a range of factors that may affect those individual freedoms, including both social factors (e.g., allocation of resources; social norms), and individual attributes (e.g., health literacy) [28]. Health in this context is defined broadly and across multiple domains of wellbeing, largely consistent with the World Health Organisation’s [30] definition of health as relating to a broad state of biopsychosocial wellbeing, and not to the absence of disease [29]. This contrasts with the guidance from NHS England suggesting that LGBT identity should only be addressed in relation to diagnosis or treatment [17].

One of the suggested strengths of the capabilities approach is that it avoids cultural relativism or individualism by seeking to identify broad domains of capability that are globally applicable, while recognising that the ways in which individuals may (or may not) choose to actually use their capabilities is governed by both cultural context and personal choice [29]. As such, the capabilities approach has been taken up by the United Nations as an approach that allows for the measurement and comparison of different countries’ actions to promote human development, without imposing prescriptive policy requirements [31,32].

A challenge with the capabilities approach and its application to health is defining the core list of capabilities [27,28]. However, Nussbaum lists 10 core capabilities that she considers governments have an obligation to provide for their citizens (Table 1) [33].

There are a number of appealing dimensions to applying the capabilities approach in considering health inequalities for minority populations. First and foremost, it allows for the possibility of different values and choices. This allows us to address issues of changes in social norms, which particularly affect older LGBT people. In the UK, the population over 60 have lived through a shift from criminalisation and pathologisation, to specific equality protections under the law, and these may continue to affect decisions around disclosure. Allowing for different values and choices also allows for application internationally For instance, we can recognise the importance of furthering individuals’ capability to engage in emotional and sexual relationships without being prescriptive about the form such relationships should take. In that sense, it aligns closely with a human rights framework, and indeed Nussbaum often refers to human rights principles in describing capabilities. The capabilities framework also allows for recognition of structural barriers to achieving health: for example that constrained choice, lack of information or poverty may affect an individual’s capability to achieve good health, even within settings that ostensibly advocate choice [28].

Ruger suggests that another key aspect of a health capabilities approach, and a key distinction from accounts focused upon individual choice, is the idea of a fundamental social obligation to promote health capabilities of everyone in society: for instance that an individual should be expected to contribute to public health services through taxation, even if they choose not to use them. Following on from that principle, public or social health initiatives should be assessed on the basis of their impact on the health capabilities of individuals [28]. This does not fully resolve the tension between utilitarianism and individualism, however, since it does not answer dilemmas where an individual could be asked or required to take action that would harm their own health capabilities, in order to benefit the health capabilities of others. One obvious example is public health restrictions introduced during the Covid-19 pandemic. More directly related to the theme of this paper is whether an individual can be asked or required to provide personal data about their own sexual orientation or gender identity, in a situation where this might harm their capabilities (e.g., through causing them emotional distress), in order to develop a dataset that can be used to further the health capabilities of the LGBT population.

While a capabilities model does not resolve all dilemmas about the collection and use of information about LGBT identities in healthcare, it does provide a starting place for exploring and expanding understandings of the relevance of LGBT identity and disclosure within healthcare. Fundamentally, it suggests that the key principle in exploring disclosures of LGBT identity in healthcare should be to seek to understand how such disclosures do or do not further the holistic health capabilities of LGBT individuals. This paper explores and developed accounts of disclosing LGBT status within healthcare by drawing upon Nussbaum’s capabilities as a theoretical framework for analysis, with a view to understanding what LGBT individuals consider disclosure allows them to do.

## 2. Methodology

This paper draws on data from a study collected for the lead author’s PhD, looking at the experiences of older LGBT people in general practice healthcare in the United Kingdom. It revisits that data to conduct a further analysis from the health capabilities perspective. Ethical approval was granted by the University of Lincoln Health and Social Care Ethics Committee on 24 March 2015. Semi-structured interviews were undertaken with LGBT people aged 60–82, living in England, Scotland and Wales in 2015–2016. Participants were recruited via LGBT community groups, social media, mailing lists, newsletters and ‘snowballing’. Initially interviews were conducted, transcribed and thematically coded in N-Vivo 10 by the lead author, using the approach to thematic analysis described by Braun and Clarke (2006), in which transcripts are transcribed, read immersively and inductively coded [34]. 

This paper is a qualitative secondary analysis, corresponding to Heaton’s categorisation of a supplementary analysis, in which additional in depth analysis is undertaken on collected data [35]. The original analysis had captured a range of issues related to experiences in general practice: this secondary analysis focused in depth on exploring accounts of reasons for disclosure. A framework approach was used for the secondary analysis [36]. The initial stages of analysis (transcription, familiarisation with data, coding) had already been undertaken within the original thematic analysis. Nussbaum’s list of 10 capabilities were then used as an analytic framework to categorise participants’ accounts of what disclosure of LGBT status in healthcare allowed them to do [33]. For example, one code that had been used in the original thematic analysis had been ‘Came out to GP because medically necessary’. This code was revisited in the secondary analysis and after close re-reading of the transcript, was mapped to Nussbaum’s capabilities of life and bodily health.

Nussbaum’s list of 10 capabilities is an established account of human capabilities. As a result, this framework was applied in a deductive fashion to explore whether participants’ accounts of disclosure could be mapped to this framework. Exemplar quotes were selected to demonstrate participants’ references to the purpose of disclosure. However, in line with Anand’s observation that not all capabilities apply in all contexts [27], ‘other species’ was rarely discussed in the context of disclosure in formal healthcare, and the capabilities of life/bodily health and affiliation/control over one’s environment were so frequently discussed together that these have been paired. The secondary analysis involved three researchers not involved in the original study (JF, THL, KA), but who have substantial prior experience of research addressing older LGBT people’s interactions with health and social care settings. Review by other analysts is a form of triangulation identified by Patton as enhancing the credibility and rigour of qualitative research [37].

In reporting findings, participants’ self-descriptions have been used. There was substantial variation in the terminology people used, and at times, personal self-descriptions did not align with suggested best practice. In particular, most participants identified either their sexual orientation or their trans status, but not both. Some participants indicated that they found standard terminology unsuitable for their needs. For example, some trans participants were uncertain how to locate themselves and their partners in regard to sexuality terminology that assumed static, binary gender. Not everyone was familiar with terms such as ‘cisgender’ to describe people who are not trans, and the notion that everyone who is not trans can be categorised as cisgender has been problematized [38]. Some participants did not describe themselves as trans, but commented that recent media discussions of gender diversity had resonated with their own experiences. Identity concepts have changed rapidly in recent years, and older people may have formed their identity within a different cultural context. This study therefore used participants’ own self-descriptions, even where this resulted a degree of inconsistency.

## 3. Results

A total of 36 participants took part in interviews (Table 2). Of the 13 participants who described themselves as trans, 9 were trans women or women of a trans history, and 11 had or intended to undergo some form of medical transition. Of the 23 participants who did not describe themselves as trans, 15 were gay men, 7 were lesbian women, and 1 was a bisexual man. The majority of participants were retired, often from public sector careers such as health or education.

### 3.1. Life and Bodily Health

Nussbaum’s first two capabilities are living to the end of one’s lifespan, and maintaining good physical health [33]. Given the previously noted emphasis on biomedical accounts of relevance, it is perhaps unsurprising that most participants indicated that they would disclose where they considered doing so was necessary to achieve either of these goals. Individuals often did not clearly distinguish between prolongation of life and maintenance of bodily health, and so these two capabilities have been grouped together.

Mark was one example of a participant who, while generally reluctant to disclose sexual orientation in healthcare, nonetheless indicated that he would do so if he considered it medically necessary:

Mark:“*I can’t think of any reason that I’ve been to see a GP about something where my sexuality would be relevant, particularly. If I’ve had a medical issue it’s never been one I can think of where that was the make or break about their diagnosis or treatment. Yet*.

Interviewer:*Would you tell them if you did think it was relevant*?”


Mark:
*“I think I probably would do for my own, sake of my own health. I’d probably have to take a very deep breath before I did it.”*
(Mark, gay man, 71)


Mark indicated that he would make a decision on a case by case basis as to whether he thought disclosing his sexuality would improve diagnosis or treatment of a presenting health complaint. In contrast, other interviewees suggested that pre-emptively disclosing sexuality could help professionals recognise health needs even if the patient did not realise there was a connection to sexuality. The examples given were primarily in relation to HIV and men’s sexual health.

“*Personally I think it’s important [to disclose sexuality to my GP] so that he knows that if there’s anything, particularly on the sexual health side*.”(Barry, gay man, 67)

“*I could not throw off a chest infection, which was unusual for me cos I was very fit and very healthy. And so I went specifically to Dr Carter and he, knowing of course that we were a partnership, a gay partnership, and recognising the symptoms as potentially a sign of HIV infection asked me if I would mind being HIV tested*.”(Jeremy, gay man, 69)

St Pierre found that lesbian women also tend to associate disclosure with reproductive and sexual health outcomes [39]. However, in this study, few lesbian women mentioned disclosing in relation to reproductive or sexual health. This may be related to the older age range.

Most accounts discussing capabilities in relation to bodily health and prolongation of life were focused on benefits to individualised processes of assessment and diagnosis. However, some participants also pointed to a potential risk of diagnostic overshadowing, again particularly in relation to sexual health. One reason John was reluctant to disclose was that an ‘out’ gay friend who was not sexually active had nonetheless been instructed to have a sexual health test when presenting with stomach problems:

“*[My friend] had a lot of problems with his stomach. It turned out to be he was allergic to something. Anyway, every time he went to the doctor—cos he used to tell me all these things—he used to say the doctor wanted him to have a sexual, you know a sexually. He said: ‘At my age, I’m afraid I don’t’. [The GP] said ‘I still want you to go to the sexual clinic’.*” (John, gay man, 68)

It is of course possible that referral to a sexual health clinic was appropriate under the circumstances. Nonetheless, John’s friend’s confusion about why he was being asked to attend a sexual health clinic, and his transmission of that confusion to John, perhaps coupled with ongoing social stigma around sexual health, led to John being concerned that disclosure could result in a reduction in his ability to access appropriate care, because a GP might inappropriately focus on sexual orientation.

While most participants focused on individual issues, some participants did point out systemic problems regarding recognition of needs:

“*Their computer systems don’t allow for, for example, monitoring of somebody who is trans. Like myself, I am now on the system as female but there are some male conditions, prostate cancer for example, that I could suffer from. And vice versa, y’know, ovarian cancer in trans guys. And I’m not sure—I don’t think for one moment that the systems allow for that. I think it’s likely that some GPs would be aware of it and would kind of put manual steps in process to cover it but I’d be very surprised if that amounted to one percent of the GPs in the country.*”(Helen, woman with trans history, 68)

While Helen identified a systemic problem within the health service in interpreting the needs of trans patients, resulting in many patients having to take on responsibility for monitoring their own care, and suggested the need for a systemic response, she was still primarily focused on the implications of applying existing knowledge to diagnosing individuals. The idea that collecting and analysing data could improve collective capabilities through generating new knowledge about the needs of LGBT populations was not raised by older LGBT participants, suggesting that, in line with wider social messaging, they did not necessarily think of the LGBT population as a group that might have distinct health needs.

### 3.2. Bodily Integrity

For trans participants, disclosure of trans status in healthcare was often directly connected to commencing the process of medical gender transition. At the time interviews took place, NHS processes in most parts of the UK required individuals to be referred by their GP to a gender identity clinic in order to receive gender-affirming interventions. Laura was relatively typical in describing disclosing her trans status to her GP in the direct context of seeking access to hormone replacement therapy (HRT).

“*[My GP] said: ‘What’s the problem?’ I said, ‘Well there’s, blah blah blah, I’m transgender, I want to, y’know, get some treatment for that, HRT, etcetera.’*”(Laura, trans woman, 60)

In contrast, LGB participants were much less likely to suggest that disclosing sexuality in general practice would contribute to their capability of achieving bodily, reproductive or sexual autonomy. In part, this may have been related to the age of participants: other studies suggest that younger lesbian women often disclose their sexual orientation in healthcare in response to the ubiquity of heteronormative messages to women on topics such as contraception and fertility [40], which may have been less of an issue for participants over 60. It may also have derived from concerns about general practice’s awareness and sensitivity to LGBT sexual autonomy: gay and bisexual male participants frequently emphasised the importance of sexual health advice and care, but reported preferring to access dedicated sexual health services rather than general practice.

Some participants highlighted that dressing and expressing themselves in their preferred way, including when they attended general practice, meant that they were automatically ‘out’ to others. This was particularly common for trans people, but was reported by some LGB participants too:

“*I used to get some funny looks [within general practice] cos I mean, I’m obviously a lesbian*.”(Smithy, lesbian woman, 65)

Conversely, some participants were concerned that, despite principles of patient confidentiality, being ‘out’ in general practice could threaten their bodily autonomy through putting them at greater risk of violence in their local communities:

“*I’ve got a partner who, we’ve been living together for, I dunno twelve years now, it is and we lived in [former home city]. And because something happened there—nothing to do with the doctor—I’m very careful about who I divulge to, and who I don’t. And we had to move because of, the police were involved and called and we were threatened and all sorts, so I’m very careful*.”(John, gay man, 68)

Ruth similarly highlighted that previous experiences of discrimination were a factor in deciding not to tell anyone—including her doctor—in her current home town that she was a lesbian:

“*I had that terrible experience in the town by the sea in [former county of residence] and I am not willing to be labelled like that*.”(Ruth, lesbian woman, 81)

Even though John and Ruth’s past experiences were not related to healthcare, the potential that disclosing in any setting could impact on their ability to live safely in their local communities was perceived to be a significant risk.

For some participants, therefore, bodily autonomy was a key factor in decisions around disclosure of LGBT+ status in healthcare. For trans people, the medicalised process for accessing interventions meant that disclosure in healthcare was a necessity to achieve bodily goals. For participants of all identities, there were considerations around the politics of visibility, with some participants knowing that their self-expression was likely to be recognised by others as LGBT, while others potentially curtailed their expression of LGBT identity in order to maximise their physical safety.

### 3.3. Senses, Imagination, Thought

Some participants reported that they disclosed their sexual orientation or gender identity in healthcare because they were newly coming to terms with their identity, and wanted to access support in understanding themselves, and/or to alleviate distressing thoughts or feelings around their identity. Julie had come out to her GP when she was experiencing symptoms of depression:

“*[Coming out to my GP] was actually in connection with how I felt at the moment, and asking her advice on that, and considering medication at that point. So when she actually said to me, asked me to actually give her some possible symptoms or at least how I felt very quickly, that was part of it, that my sexual identity was a big issue and so that’s how I told her really*.”(Julie, lesbian woman, 65)

There were mixed perceptions on how far GPs were able to provide support for thoughts and feelings around sexuality or gender identity. Some participants reported a lot of support and reassurance from their GPs:

“*I started a lot older than the average. I was sort of, nearly, I was sixty-eight when it all sort of cropped up. I really wanted that, just the chat which was just to say, yes, not sort of justification in a way but it’s that ‘Yes we think you’re doing the right thing’. I had a niggle and I thought I’d got it wrong [….] It’s that sort of that cuddly feeling in a way that’s: ‘Don’t worry everybody goes through that, that’s quite normal. Don’t panic, just take it easy, you know, you’ve got another three months to go before you see your, before you’re seen by the gender people. That’s quite normal, don’t worry about it, you know you’ll be all right when you get there’*(Eleanor, trans woman, 72)

Others were sceptical about whether medical staff could fully understand the experiences of LGBT patients, and indicated that they would prefer to access dedicated LGBT services for support around emotional and psychological wellbeing.

“*I think you need to know, that when you take a problem to them, when you take a medical problem to someone, they are going to be sympathetic. If you’ve got a system like [dedicated LGBT charity] then it, then they bloody well should be, you know. But you can’t go to, and in the same way that as if you’re, if you’re not a depressive you can’t understand, in the same way if you’re not gay you cannot*.”(Lennie, gay man, 73)

Steve suggested that GPs might not have the time or resources to engage with issues of wellbeing and lifestyle, repeatedly indicating that he didn’t know what was reasonable to expect from a GP in this regard:

“*Is there a lack of interest in lifestyle? I don’t… Should there be a greater interest in lifestyle and happiness? I don’t know. Have they got the time, interest or facilities? I don’t know*.”(Steve, gay man, 67)

These participants all emphasised that being able to access LGBT-inclusive support and reassurance enhanced their capabilities to address difficult thoughts and feelings. However, they differed in their expectations as to whether disclosing LGBT identity in general practice would achieve this.

### 3.4. Emotion

For many participants in partnered relationships, acknowledgement of the significance of their emotional bond was the key reason for coming out in healthcare. Participants often emphasised the importance to them of same-sex relationships being treated as equally emotionally significant to different-sex relationships.

“*I hate it when they say ‘He’s a married man’, cos they don’t say that about me and Peter. It’s usually ‘Peter and his friend’, you know what I mean? Sometimes, like sometimes when we go and see a doctor we don’t know […] Peter always says ‘Do you mind if me husband comes in?’, you know. Usually they might just, a quick look and then it’s over and done with, isn’t it*?”(Mike, gay man, 62)

Moira had supported her partner through terminal illness:

“*Cheryl always introduced me as her partner, so it was right. Whether it was the doctors, the nurses, or the consultants, so she was very clear. And so, no I never had any, I never found any difference*.”(Moira, lesbian woman, 68)

For Mike, Moira and their partners, introducing a same-sex partner, and emphasising that this was a partner or a spouse, and not merely a ‘friend’, was important for ensuring that medical staff understood the significance of their relationship. Lesbian and gay participants frequently described ‘coming out’ in health through referring to a partner’s gender, rather than making any specific statement about their own sexuality. However, the normatively higher value placed on partner relationships could potentially have had a detrimental impact on the capabilities of individuals to emphasise other important emotional bonds: for example, close friendships. The assumption that current partner status is synonymous with sexuality may also tend to obscure the identities of bisexual people and those not currently in relationships. Among participants in this study, several had previously been in different-sex marriages, and use of the title ‘Mrs’, or the recording of marital status as ‘widowed’ or ‘divorced’, may have resulted in an assumption of heterosexuality within healthcare.

### 3.5. Practical Reason

Under practical reason, Nussbaum includes ability to plan for the future, and ability to reach one’s own decisions on morality [33]. In healthcare, these issues intersect with structural issues of power in the professional-patient relationship, and the extent to which patients have the freedom to assert their own plans and decisions.

The emotional significance of recognising key relationships addressed in the previous section interacted closely with the question of planning for the future: recognition of spouses, partners and ‘next-of-kin’ was associated with issues such as who could take decisions in an emergency. The notion of “next-of-kin” is not one formally set in law, but is often interpreted in terms of spouses or biological relatives [41]. For example, Mike explained that one of the reasons why he and Peter emphasised their marriage was because many years earlier in their relationship they had not being permitted to visit each other in hospital:

“*Because he’d been married, his ex-wife could go in and see him in hospital and yet me, we’d been together by that time, say, twelve years, by law I’d got no rights. His ex-wife came first, even though they’d divorced and Peter had to write to a Member of Parliament and the Prime Minister. It took a long time and a lot of hassle to get them to say that I was Peter’s next of kin and he was mine*.”(Mike, 62)

Jeremy and his partner had made a point of ensuring that they were registered as a couple.

“*We went along to register as a couple. I made it very clear to them that this was the case*.”(Jeremy, gay man, 69)

For trans participants, disclosure in general practice was often also a practical necessity in terms of ensuring that health records used the correct name and title. In addition, trans people require a doctor’s letter to update their UK passports, and several participants mentioned asking their GP to write this letter.

Nussbaum also includes under practical reason the capability to reach one’s own decisions about morality. Several participants described situations where they or people they knew had encountered medical professionals who appeared to not approve of their sexuality or gender identity, ranging from historic experiences of conversion therapy, to more recent experiences of staff who seemed rude or dismissive:

“*In the fifties, you know, I went to the doctors when I was sixteen and I said to the doctor, I said ‘Well how is it I like men more than what I like women?’ Because I did, you know, and ‘Ooh’ he said ‘Well you should be, you should like women’ and that’s when I went on this gay cure thing*.”(Pete, gay man, 74)

“*Well, there’s a nurse that is well known to be homophobic at the surgery and I’ve got plenty of friends who, you know, none of us want to see her*.”(Zenobia, lesbian woman, 64)

There is of course a difference in scope between conversion therapy in the 1950s and a nurse who seems prejudiced to her lesbian patients today. There have been significant intervening changes in UK law, with the Equality Act (2010) prohibiting discrimination in service provision. Most participants indicated that they had not recently experienced overt discrimination in healthcare, nor did they expect to. However, some were concerned that staff might be prejudiced but not express it. Some also expressed particular concern about health practitioners with religious beliefs. The power imbalance in the patient-medical practitioner relationship meant that a professional’s beliefs potentially could affect access to care, resulting in reduced capability to access care, or a sense of vulnerability when asking for care.

### 3.6. Affiliation/Control over One’s Environment

Under affiliation, Nussbaum highlights the capability to participate equitably in society, free from discrimination, as well as being able to engage in social groups and social interactions. The capability for control over one’s environment encompasses political participation. When it came to discussing the implications of disclosing LGBT status in healthcare, the concepts of equitable social participation and political participation were closely linked and so these two capabilities have been considered together.

A small number of participants (predominantly those living in areas with a large, well-established LGBT community) described initiatives specifically intended to encourage LGBT people to provide feedback and engage with development of health services. In one case, this extended to inviting patients to discuss key concerns with new members of staff:

“*When they’re [healthcare staff] replaced, the community has an opportunity to have a meeting where we offer input. They send out a shout-out by email saying: ‘Do you want to come along and discuss anything in particular like, you know older people’s concerns, autism, transgenderism or whatever’ and they really do take on board what you say. Which is remarkable really*.”(Frankie, non-binary person, 62)

There were mixed views on other approaches to addressing discrimination and engaging LGBT communities. Some participants were strongly supportive of initiatives such as rainbow flags and accreditation schemes:

“*I think certainly if you saw [a rainbow flag] in their window you’d think ‘Oh yes, go in there no problem. I think particularly [certification schemes]. These are just symbols, aren’t they, but an award, if you like, would be an excellent idea*.”(Laura, trans woman, 60)

Interviewer:“*Do you think there’s anything more doctors could do to make it easier for people to come out to them*?”


Pete:“*Well, yes they could because they could put a rainbow flag in the, in the waiting room or—well a rainbow flag is sufficient*”(Pete, gay man, 74)


Others expressed scepticism about how effective equality measures were in practice.

“*People never really take much notice of [equality policies] any more. They just pay lip service to it. So I don’t think it would really mean anything. [….] I mean it’s important to have one in place, yes, because you know somebody might take it up and say ‘yes, I have definitely been discriminated against on the grounds of sexual orientation or gender identity or age or both’, you know. So it’s important to have it there but I think it needs to have a higher profile*.”(Smithy, lesbian woman, 65)

“*It’s easy to say ‘We’re equal opportunity, we’re equal opportunities employer’ and so on. Grass roots, my experience is actually, it doesn’t really mean a great deal*.”(Jan, lesbian woman, 60)

Participants tended to suggest increasing LGBT awareness within healthcare took place within the broader context of social change:

“*I think it’s just an evolutionary thing and things will improve. As I say, I’ve seen them come on leaps and bounds. It’s not perfect by any stretch of the imagination but considering what it was like in 1960, it’s a paradise*.”(Oscar, gay man, 65)

“*I think they’ve [doctors] got the same, you know, the same cultural prejudices and shyness about it, awkwardness about it, lack of awareness about it as the general population really. I mean, hopefully they should have done some modules with training. I think younger clinicians are probably better*.”(Margaret, lesbian woman, 67)

Relatively few participants recalled coming across attempts to monitor the experiences of LGBT patients in healthcare, and those that did seemed similarly uncertain how much difference it made in practice:

“*I recognise its importance, but there’s some aspects of it I feel slightly—not necessarily intrusive but I wonder what’s behind the question. How it’s being used. And sometimes I have to admit I wonder if it’s just done for the sake of it and it’s just all filed away, you know, to a cardboard file and forgotten about*.”(Sophie, lesbian trans woman, 63)

Overall, participants emphasised that they had witnessed substantial improvements in attitudes to LGBT people within healthcare, but tended to see these as deriving from organic processes of social change. Some equalities initiatives were perceived as being rather tokenistic. However, as with Frankie’s experiences above, those participants who had encountered meaningful and positive attempts to engage with LGBT communities were extremely supportive of such initiatives.

The research study was primarily focused upon primary care practice, as in the UK this is the main provider of preventative and ongoing care, and is the usual referral route for secondary care. However, several participants spontaneously raised the issue of later life care and their perceptions of the likelihood (or otherwise) that if their health declined, they would be able to continue to live in a dignified manner, in an environment they felt comfortable in, while being openly LGBT. Several indicated that they would prefer to seek out LGBT-specific retirement housing, although they were aware that relatively few options were available. Chris and Jan described concerns based on older lesbian and gay people they knew.

“*They [gay male friends] get to a point where they’re no longer able to live on their own and a lot of them, a big percentage of them, are having to go back into that closet that it took them years to get out of. Because by and large the people within their age group is not as accepting of gay people as the younger generation and the percentage of people in care homes, the majority of them are women rather than men and therefore they have to fit in to that little group. And there is no alternative*.”(Chris, gay man, 66)

Interviewer:“*What do you think health services or GPs could do to sort of start addressing some of these [issues]*?”


Jan:“*Maybe trying to put some pressure on local authorities to have—maybe not homes but, well, yeah—which gay people could go to, or having the option to go to. I think for that generation, the generation above mine, how hard would it be for them to come out*?”(Jan, lesbian woman, 60)


The issue of disclosure therefore potentially became much more significant when participants considered a potential future living in a social care setting. As reported in other studies, several participants saw care homes as a setting where they would not be able to exercise control over their environment, would not be treated with dignity if they were openly LGBT and would be expected to live according to heteronormative expectations [42,43].

### 3.7. Play

Under play, Nussbaum lists the ability to laugh and to engage in recreational activities [33]. There were few references to disclosure of LGBT status in healthcare facilitating capability to engage in recreational activity, except in the context of sexual health disclosures, addressed in previous sections. Similar to the issue of ‘other species’, while several participants spoke about the personal wellbeing benefits of arts and leisure activities, they did not seem to see much connection between these activities and use of healthcare services. This may point to the ongoing prioritisation of a biomedical understanding of health.

Some participants did highlight the benefits of friendly, informal and jokey interactions with practice staff, and that this sometimes incorporated affirmation of LGBT identity, especially in regard to recognising partner relationships:

“*It’s quite like a family, you know. If Jackie and I go in, they’ll say ‘Oh, hello you two, are you all right?’ ‘No we’re not, that’s why we’re here*.’”(Frankie, non-binary person, 61)

“*We walk in our surgery, you know, if I go on me own [to] pick up a tablet they say ‘Where is he? Have you had a row*?’”(Mike, gay man, 62)

In practice, there were mixed experiences as to whether participants were able to develop these kinds of informal relationships with healthcare staff, in part related to continuity of care. Those who had managed to develop more personal relationships with staff clearly valued them.

## 4. Discussion

This paper has applied a capabilities approach to analysing LGBT people’s accounts of disclosing sexuality and gender identity in healthcare. LGBT health research is at times under-theorised, and applying the health capabilities model to the disclosure decisions of a diverse sample of older LGBT people provides an additional theoretical basis for understanding the links between disclosure and health. Focusing on how disclosure may enhance (or inhibit) capabilities develops the analysis away from simply establishing what is considered to be ‘relevant’, and instead addresses what disclosure of LGBT status allows LGBT people to achieve within healthcare. It also provides context for exploring different mechanisms LGBT people may use to enhance their health capabilities: for example that some individuals who want options to address difficult feelings about sexuality may do so through talking to their GP, while others may seek out dedicated LGBT services. In the context of this study, which focused on state-provided primary care services, some capabilities identified by Nussbaum (e.g., play and other species) were not frequently raised by participants. It is possible that participants may have addressed these health capabilities in other ways, e.g., through community organisations.

The use of health capabilities in this context is novel, but close parallels can be drawn with other literature exploring disclosure in healthcare, which frequently highlight the significance of questions around the relevance and purpose of disclosure [11,19,20], and the balancing act older LGBT people may face in weighing up the pros and cons of disclosure in different settings [42,44,45,46]. Past analysis has also emphasized that the health and wellbeing benefits of disclosing LGBT status go beyond purely biomedical conceptualisations of health [18,23]. This analysis develops such accounts by suggesting that the health capabilities model can be used as a conceptual framework to map the range of purposes that disclosure may serve in enhancing health.

Capabilities tended to be framed in a personal context: participants considered what disclosure within their own healthcare interactions would allow them to do individually, rather than considering social dimensions of capabilities in terms of implications of disclosure for the LGBT population as a whole. This focus on individual interactions is perhaps unsurprising in the context of a study exploring experiences in general practice. It also to some extent parallels the NHS guidance that recording of sexual orientation should be linked to implications for individual treatment [17], although participants did not solely focus on treatment, and instead pointed to a more holistic range of capabilities that they sought to achieve, such as recognition of key emotional relationships and access to support. Some participants also highlighted concerns that disclosure of LGBT status could pose a threat to their individual capabilities, for example through exposing them to risk of discrimination or violence, or increasing the likelihood of receiving poor care.

There were some indications that participants’ identities and experiences affected how they weighted the impact of disclosure upon their personal capabilities. Trans participants were more directly dependent on disclosing their trans status in order to access to gender affirming interventions, in part due to the structures of trans healthcare in the UK [38,47]. Individuals in long-term relationships often primarily emphasised the relationship, rather than sexual orientation. Sexually active men tended to focus on the implications for recognising and diagnosing sexual health issues, in line with previous observations that LGBT health is often interpreted as being primarily concerned with sexual health [11]. Other participants—in particular those who were single, not trans, and not currently sexually active—simply could not think of any benefits that disclosure would achieve. The fact that disclosure of LGBT status fulfilled different capabilities, and that some capabilities were more relevant to some groups of participants than others, suggests that it is likely there will be differences in who discloses under what circumstances, and that individuals may change their decisions on disclosure according to context.

The public health case for disclosure of LGBT status in healthcare, in contrast, rests primarily on population-level benefits to disclosure: that collecting data on the whole LGBT population could improve the capabilities of LGBT people to achieve good health, for example through better understandings of risk, and targeting of interventions [5]. These collective capabilities are most enhanced if all LGBT people disclose their status. Disclosure based on individual impact on capabilities is likely to underrepresent those parts of the population whose capabilities are least enhanced through disclosure, or who see most threat to their capabilities from disclosure. These may be the LGBT people who experience the greatest health inequalities.

Given that the focus of the capabilities approach is on creating social circumstances that give individuals freedom to make choices to improve wellbeing [33], it would be inappropriate (as well as practically unworkable) to require individuals to disclose. Several participants in this study had genuine and justifiable concerns about the implications of disclosure, often rooted in prior experiences of discrimination. For individuals in this situation, removal of the perceived threat to capabilities arising from disclosure may be the most important factor. Paradoxically, however, participants who are concerned about individual threat cannot be targeted individually, because health services do not know who they are. Therefore, addressing these barriers requires collective messages on issues such as confidentiality and non-discrimination, to minimise perceived threats.

For other individuals, non-disclosure was more closely related to a neutral impact on individual capabilities—they did not feel that disclosing LGBT status in healthcare would make much difference one way or another to the options open to them as individuals. For some of these individuals, a clear and credible message that disclosure could enhance collective health capabilities for the LGBT community might be persuasive. Many participants in this study were active in LGBT communities and concerned for the wellbeing of others, and might be willing to disclose their identity if convinced that doing so would help to improve the health capabilities for other people. Since this research took place, variations on the slogan “If you don’t count us, we don’t count” have been increasingly apparent in UK LGBT discourse, particularly in communications by the UK’s national advisor for LGBT health [48,49]. This seems to be an appeal to a notion of collective capability: a call on behalf of the LGBT community, to healthcare providers, in order to establish that “we count”.

One potential hindrance to appeals to collective capability would be the apparent scepticism observed among study participants regarding the practical outcomes of equality initiatives. For example, Sophie, quoted earlier, indicated that she was aware of the arguments in favour of monitoring, but was uncertain whether data was actually being used. Similarly, equality policies and training were frequently seen by participants as tokenistic or believed to be given low priority. This suggests that there may be particular benefits to consulting with LGBT communities on research gaps, research priorities and enhancing dissemination of health research back to LGBT communities, and maintaining such conversations over time. Not only could this potentially enhance capabilities through giving LGBT people and communities information that they could use in understanding and advocating for their own needs, but it could also enhance future data collection through helping to convince LGBT people—and healthcare providers—that disclosure has value. 

The philosophy of the capabilities approach is, however, focused on the social level, emphasising the need for policy makers to create the circumstances in which individuals develop holistic capabilities needed to achieve a good life [33]. As such, understanding and improving LGBT health cannot be seen as the sole responsibility of LGBT people themselves, nor of researchers. Establishing standards for collecting data is an important initial step in providing the framework to collect data, but the larger project is to create a social context in which older LGBT people can feel certain that talking openly about their sexual orientation and gender identity will enhance rather than limit their health capabilities.

### Limitations

This paper is a secondary analysis of data originally collected for an exploration of older LGBT people’s experiences in UK general practice care. Participants were recruited via LGBT support organisations, and in common with other studies, certain groups (bisexual people, LGBT people of colour, LGBT people over 80) were poorly represented. This paper indicates that capabilities are a potentially useful framework for exploring decisions around disclosure of LGBT status in healthcare, but assessments of capability are likely to be highly contextual. For example, research on LGBT health capabilities in other settings might give greater weight to some capabilities (other species, play) than was raised by participants commenting on state-provided primary care provision.

## 5. Conclusions

Disclosures of LGBT status by older people can further a number of different aims, from perceived direct medical benefits, to recognition of emotional bonds, to being able to pursue bodily autonomy. These aims do not necessarily always align with either biomedical accounts of disclosure (e.g., to inform diagnosis or treatment), or public health accounts of disclosure (to identify and address population level inequalities). For some individuals, disclosing can also be a threat to capability. Even in the context of the UK, where public bodies are mandated to promote equality and avoid discrimination, some individuals still experienced or feared the possibility that disclosing LGBT status would negatively affect their health capabilities. These perceived threats may be greater in contexts where there are fewer protections for LGBT individuals. Explicitly recognising the different ways in which disclosure can enhance or reduce capability, and the contextual factors affecting that, may help to explore local and global barriers to collecting data on LGBT populations, as well as providing opportunities to improve the quality of data through removing or reducing threats.

In situations where it appears that the primary barrier to disclosure is not threat but a perceived lack of relevance to individual care, direct appeals to collective capabilities—‘what does disclosure allow us to do’—may be a potentially fruitful approach to encouraging disclosure of LGBT identity in healthcare. Further research could usefully explore the extent to which collective capabilities are indeed a beneficial message, and how this could be promoted by researchers, third sector organisations, healthcare practitioners and government bodies.

## Figures and Tables

**Table 1 ijerph-17-07614-t001:** Nussbaum’s 10 central human capabilities.

Life	Being Able to Live to the End of One’s Lifespan without Premature Death
Bodily health	Being in good physical health, including reproductive health
Bodily integrity	Being able to move freely, being free from violence, having bodily, reproductive, and sexual autonomy
Senses, imagination, thought	Being able to reason, think and create, access to scholarly traditions such as art, literature and science. Having pleasurable experiences and avoiding non-beneficial pain
Emotions	Being able to form and mourn emotional attachments to others
Practical reason	Being able to conceptualise what is good, and being able to plan one’s future
Affiliation	Subdivided into interactions with others, and dignified non-discriminatory participation in society
Other species	Being able to live with concern for animals, plants and the natural world
Play	Laughter, play, recreational activities
Control over one’s environment	Subdivided into political participation and material rights to own property and undertake employment

Adapted from Nussbaum [33].

**Table 2 ijerph-17-07614-t002:** Participant Information.

Pseudonym	Identity	Age	Work Status
Steve	Gay man	67	Retired
Sandra	Tranvestite	69	Retired
Vonni	Trans woman	68	Retired
Laura	Trans woman	60	Own business
Smithy	Lesbian woman	65	Own business
David	Gay man	69	Freelance, part time
Helen	Woman with trans history	68	Retired
Caroline	Bi trans woman	67	Retired
Anne	Trans woman	82	Retired
Liz	Trans woman	67	Retired
Zenobia	Lesbian woman	64	Retired
Ralph	Gay man	67	Retired
Julie	Lesbian woman	65	Retired, with occasional freelance work
Maurice	Gay man	74	Retired
Margaret	Lesbian woman	67	Retired
Barry	Gay man	67	Retired
Sophie	Lesbian trans woman	63	Retired
Robert	Gay man	72	Retired
Jeremy	Gay man	69	Retired
Dee	Trans man	75	Retired
Oscar	Gay man	65	Employed
Lennie	Gay man	73	Retired
Mark	Gay man	71	Retired
Pete	Gay man	74	Retired
Mike	Gay man	62	Retired
Strich	Gay man	82	Retired
Moira	Lesbian woman	68	Retired
James	Trans man	64	Retired
Chris	Gay man	66	Retired, trustee of charity
John	Gay man	68	Retired
Archibald	Bisexual man	64	Retired
Ruth	Lesbian woman	81	Retired
Eleanor	Trans woman	72	Retired
Jan	Lesbian woman	60	Employed
Elizabeth	Trans woman	62	Unemployed
Frankie	Trans non-binary person	62	Part time

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
