# Peer review of "Applying a Capabilities Approach to Understanding Older LGBT People’s Disclosures of Identity in Community Primary Care"

_ijerph, 2020, doi:10.3390/ijerph17207614_

Round 1

Reviewer 1 Report

The paper is of interest and addresses older LGBT individuals whose experiences are more pertinent with current healthcare practices. The authors present a sound thesis throughout, yet, some methodological queries remain. This is possibly as this is a qualitative secondary analysis (QSA), which the authors should be claiming a s methodology.

Also, further details about how thematic analysis was applied would help the reader understand how the emerging themes were aligned to the capabilities framework. Alternatively, as the data were made to fit the framework, this does not seem to be following a thematic analysis.

Author Response

We thank reviewer 1 for their overall positive comments on the paper.

As the reviewer correctly notes, this was a secondary analysis, and we have included additional text in the methodology section addressing the secondary analysis in more depth.

Reviewer 2 Report

There are three concerns, which could make the paper much stronger.

  1. You cite Braun and Clark, researchers who popularized thematic analysis. Yet, you do not actually mention the steps. Did you merely recreate their six steps? How did you do that, as this study is deductive? How did you code? Categorize? Subtheme? Theme? Substantive theory?
  2. You look at health capabilities, but under Section 3, I feel codes, subthemes, and themes, including memos and other analysis, are missing. Without this, while the writing is exemplary, there seems to be a wandering within the sections. A reader cannot clearly establish descriptions and codes evolved into categories and themes beyond the anecdotal. Also, and importantly, were some themes more important and clearer than others? Right now, I feel like maybe all were equally important, meaning nothing was important. What stood out? Example: 353-364 about next-of-kin makes this concern real; there is one example and no other subthemes or issues. If I had to rank the sections, 3.7 was the best, followed by 3.1. For example, 217-222 gives some descriptions of categories under 3.1. I guess I want a more systematic approach; you stated you used a systematic approach, but you quote entire sentences and show no codes or smaller units. Two more major questions: Was there a substantive theory which bound all themes together? Were there other capabilities or an absence, as undoubtedly Nussabaum might not have captured all capabilities?
  3. Validity and reliability in qualitative research are important. How did you determine trustworthiness? Did you check dependability, credibility, transferability, and confirmability? Triangulation, such as data analysis, researcher, theoretical, and methodological? There are other schools of thoughts, but the point is there should be attempts to say the themes you found were most important, fit the data, usable by other practitioners, and others would find the same or similar results if they conducted the study. Without explaining why your information is trustworthy, and often only seeing one example for claims with no codes, descriptors, or subthemes, there is doubt the entire sample pointed to the themes described versus a haphazard, all over the map ambiguity which did not produce a pattern.

Typo 188 “twocapabilities”

Typo 203 “take” should be “make”

Author Response

We thank reviewer 2 for their detailed comments.

  1. Additional detail on the methodology has been provided. As reviewer 1 pointed out, it should have been made clearer that this was a qualitative secondary analysis of data that had originally been thematically coded, using Nussbaum’s list of capabilities as a framework. A fuller description of the process has been provided, including discussion of how codes from the initial thematic analysis were used in the secondary framework analysis.
  2. This set of comments to some extent follows on from the earlier point about clarifying methodology. Nussbaum’s list of health capabilities was used as a deductive framework to structure the analysis, and is the theory that underpins the paper. This has been made clearer in the text. The sequencing therefore follows Nussbaum’s list.
    As Nussbaum’s list of capabilities are broad, therefore the data under each section does cover a diverse range of issues. Part of the argument of this paper is that older LGBT people reference a diverse range of health capabilities in discussing their reasons for disclosure, and that the capabilities framework could conceptually be applied to understanding different reasons for disclosure without necessarily privileging certain reasons over others (in contrast to existing perspectives in health systems, which do tend to focus on biomedical reasons for disclosure). We have more clearly highlighted this in the discussion section, along with noting that some capabilities identified by Nussbaum were not frequently raised by participants in the context of formal primary care.
    Additional supporting quotations have been provided at various points within the findings.
  1. As the reviewer notes, there are a number of different perspectives on ensuring rigour and credibility in qualitative research. Guba and Lincoln (1985) stress the importance of establishing trustworthiness in qualitative research through the qualities credibility, transferability, dependability and confirmability, with an emphasis on reflexive practice, “thick description” and discussion of the choices taken within the research process. We hope that we have further enhanced these qualities within the revised version paper through additional explanation of the research methodology (including the benefits of review by multiple authors for analytic transferability); additional text in the discussion addressing the extent to which the findings of this study are comparable to other research findings; and additional discussion of the theoretical contribution of this paper to wider debates.

Some minor typos identified by this reviewer have been corrected.

Reviewer 3 Report

GENERAL COMMENTS

The purpose of the paper was to interview LGBT individuals in the UK about healthcare, and apply their responses to the health capabilities framework. It is a creative and well-written paper, on a topic that has not been frequently studied. It is rare that I read a paper and only find a few things to correct, and this is one of those papers!  The authors have done a fantastic job writing this paper. 

I have two minor suggestions that should be addressed prior to publication, which I have written in the "Specific Comments" section below.

SPECIFIC COMMENTS

p. 6 of 15, lines 224-225: The quote was about "the doctor wanted him to have a sexual, you know a sexually....." (It seems like this sentence was incomplete as the next sentence mentioned what seems to be a STD clinic.)  Check this quote to make sure it is correct.

p. 11-12 of 15 (Discussion): This discussion needs a bit of work. The majority of the section re-iterates findings. I would like to see a more thorough discussion of how these results compare to other studies--even if from other countries (if applicable), and if other comparative studies are not available, the authors should state that explicitly. In addition, I'd like to see a more thorough discussion of the implications of some of these findings.

Author Response

We thank reviewer 3 for their extremely positive comments about the paper.

Regarding their two suggestions:

The participant quote at line 225 is correct: the participant did not complete his sentence after the word ‘sexually’. However, the end of the quote does make it clear that what was being discussed was a referral to a sexual health clinic. We have slightly amended the following text so that the reference to a sexual health clinic is consistent throughout.

Additional text has been included within the discussion. This more clearly addresses how the findings of this study relate to previous research on disclosure of LGBT status in health and social care services, and how applying a health capabilities approach develops and extends the literature.